# The Influence of Socio-Demographic Factors, Lifestyle and Psychiatric Indicators on Adherence to Treatment of Patients with Rheumatoid Arthritis: A Cross-Sectional Study

**DOI:** 10.3390/medicina56040178

**Published:** 2020-04-14

**Authors:** Adina Turcu-Stiolica, Mihaela-Simona Subtirelu, Paulina Lucia Ciurea, Dinescu Stefan Cristian, Maria Bogdan, Andreea Lili Barbulescu, Daniela-Gabriela Glavan, Razvan-Aurelian Turcu-Stiolica, Sineta Cristina Firulescu, Beatrice Andreea Chisalau, Cristina Dorina Parvanescu, Bogdan-Petre Stanoiu, Andreea Daniela Meca, Johny Neamtu, Florentin-Ananu Vreju

**Affiliations:** 1Department of Pharmacoeconomics, University of Medicine and Pharmacy of Craiova, 200349 Craiova, Romania; 2Department of Rheumatology, University of Medicine and Pharmacy of Craiova, 200349 Craiova, Romania; ciureapaulina@yahoo.com (P.L.C.); stefandinescu@yahoo.com (D.S.C.); florin_vreju@yahoo.com (F.-A.V.); 3Department of Pharmacology, University of Medicine and Pharmacy of Craiova, 200349 Craiova, Romania; bogdanfmaria81@yahoo.com (M.B.); anbarbulescu@gmail.com (A.L.B.); andreea_mdc@yahoo.com (A.D.M.); 4Department of Psychiatry, University of Medicine and Pharmacy of Craiova, 200349 Craiova, Romania; danaglavan@gmail.com; 5Trueman Consulting, 200528 Craiova, Romania; razvan.turcu@gmail.com; 6Doctoral School, University of Medicine and Pharmacy of Craiova, 200349 Craiova, Romania; sineta.firulescu@gmail.com (S.C.F.); beatrice_med@yahoo.com (B.A.C.); parvanescu.reumatologie@gmail.com (C.D.P.); 7Department of Cell and Molecular Biology, University of Medicine and Pharmacy of Craiova, 200349 Craiova, Romania; stanoiu.bogdan@yahoo.com; 8Department of Pharmacy I, University of Medicine and Pharmacy of Craiova, 200349 Craiova, Romania; j.neamtu@yahoo.com

**Keywords:** lifestyle, socio-demographic factors, rheumatoid arthritis, treatment adherence, CQR-9, PDSQ, major depressive disorder

## Abstract

*Background and Objectives*: Rheumatoid arthritis (RA) is a severe autoimmune disease characterized by chronic inflammation of the joints accompanied by the progressive deformation and destruction of cartilage and joint bones. This study aims to gain insight into the outcomes related to adherence in patients with rheumatoid arthritis. Predicting the medication adherence in RA patients is a key point to improve the treatment outcome. *Materials and Methods*: A number of 119 Romanian patients with RA were included and divided into two groups: first group included 79 patients treated with conventional therapy and second group included 40 patients treated with biologic therapy. A CQR-9 (compliance questionnaire rheumatology with nine items) and PDSQ (psychiatric diagnostic screening questionnaire) were performed to assess correlations between medication adherence, patient sociodemographic variables, 11 psychiatric scales (major depressive disorder, posttraumatic stress disorder, obsessive-compulsive disorder, panic disorder, psychosis, agoraphobia, social phobia, drug abuse/dependence, generalized anxiety disorder, somatization disorder, hypochondriasis) and lifestyle (bulimia, alcohol intake). *Results*: Whilst modelling factors associated with adherence, it was found that women and patients with higher education are more adherent. From the psychiatric indicators, only major depressive disorder and post-traumatic stress disorder were found to be positively correlated with therapeutic adherence. None of the assessed lifestyle factors influenced the adherence of RA patients. *Conclusion*: The knowledge of factors that impact on treatment adherence can be useful for clinicians to guide patient-centred care.

## 1. Introduction

There are more than 100 rheumatic diseases afflicting joints and surrounding tissues, rheumatoid arthritis (RA) being one common rheumatic condition [1]. The management of RA was dramatically shifted towards new treatments such as biologics and anti-tumor necrosis factors (TNFs) [2]. Despite new pharmacological strategies, patients with RA still report unmet needs due to symptoms that remain unaddressed [3]. Some lifestyle factors (bulimia, cigarette smoking, alcohol intake, caffeine and sugar-sweetened soda consumption) were reviewed as they may affect RA risk [4] and some (dietary interventions) were investigated to verify whether they improve RA symptoms [5]. Alcohol intake for RA patients treated with methotrexate and leflunomide, that are potentially hepatotoxic, could increase the risk of cirrhosis. Bergman [6] found an association between alcohol consumption and both lower self-reported disease activity and higher quality of life in female, not in male, RA patients, without assessing the differences between conventional and biologic treatment.

Adherence was defined by the World Health Organization as: “The extent to which the patient’s behaviour matches agreed recommendations from the prescriber” [7]. This varies from the aged term “compliance” by emphasising the need for agreement. Comorbid mental health problems in patients with RA are associated with reduced adherence to therapy, as well as remission of RA symptoms, impaired quality of life, increased disability and mortality and enlarged health care costs [8,9]. It is considered that enhanced medication adherence could be even more important on health outcomes than advances in medical therapy [10], with a less significant cost burden on healthcare systems [11].

Very few studies have shown correlations between lifestyle factors and adherence in RA patients. Alcohol drinkers vs. non-alcohol drinkers showed good adherence with pregabalin (81% vs. 61%) and tramadol/acetaminophen (88% vs. 65%) in RA patients [12].

In rheumatic diseases, therapeutic adherence varies a lot, depending on different variables [13,14,15] including age, type of treatment, duration of disease or medication beliefs [16,17,18,19].

Medication adherence could be measured using several methods, including self-report questionnaires, electronic devices, pick-up/refill prescriptions, or a drug’s quantification in patient serum (using a liquid chromatographic method [20]) or other sampling matrices (e.g., hair, saliva and urine). Although not the most exact method, patient-reported outcomes, such as self-reported questionnaires, are the easiest to find reasons why the patients have low medication adherence.

Depression may negatively impact memory, planning ability, beliefs about therapy and efficacy. However, it is a modifiable factor if it is correctly diagnosed and treated. Therefore, the risk for a diminished adherence can be alleviated [21].

RA also has extra-articular impacts, like depression and anxiety. The prevalence of depression is approximately double compared to anxiety, estimated to be 38% [8,9]. Depression and anxiety have similar causative factors such as pain, inflammation, and disability, and are frequently found together in patients [8].

Depression occurs more often in RA patients than in the general population [22], patients with RA being 2–3 times more likely to have major depression than individuals without rheumatic conditions [23,24].

Depression complicates the management of other conditions (severe trauma, cancer, diabetes, myocardial infarction) or can occur secondary to other diseases like inflammatory conditions, Parkinson’s disease and hypothyroidism [25]. Inflammatory processes play a key role in the pathogenesis of depression. Several disorders with an inflammatory basis, including RA, multiple sclerosis, inflammatory bowel disease, cardiovascular disease, chronic liver disease, cancer, and diabetes [23,26], often co-occur with depression. Multiple proinflammatory markers (like Interleukin (IL)-1, IL-6, tumour necrosis factor alpha (TNF-α), C-reactive protein - CRP) have elevated values in depressed patients compared to nondepressed individuals [23,24]. Increasing evidence emphasizes that systemic inflammation could provoke depression by inducing neuroinflammation [26].

The Diagnostic and Statistical Manual of Mental Disorders, Fifth Edition (DSM-V) from the American Psychiatric Association is the golden standard for diagnosing during a psychiatric interview, yet expensive and time-consuming. Other tools that can be used are self-report screening questionnaires which are efficient, quick and easy to manipulate [27]. The hospital anxiety and depression scale (HADS) and patient health questionnaire (PHQ) are typically screening instruments [23]. A Romanian adapted screening questionnaire is PDSQ (psychiatric diagnostic screening questionnaire), which consists of 13 subscales and provides information from six areas: mood, eating, anxiety, substance use and somatoform disorders, and psychosis [28].

There is not much literature on the influence of lifestyle factors and mental health symptoms on RA medication adherence. This study aimed to find potential correlations between the level of adherence of patients diagnosed with RA and several outpatient mental health disturbances or their lifestyle, in order to propose a model for predicting adherence to therapy in RA. We studied also the differences in terms of adherence between the biologic and synthetic Disease-modifying antirheumatic drug (DMARD) therapies.

## 2. Materials and Methods

The current study is a cross-sectional patient survey study conducted in Romania between December 2017 and February 2019. Adherence was measured with the CQR-9 questionnaire (translated, adapted and optimized through statistical methods [26,27], see Table 1). Medication adherence was expressed as a percentage value whereby 0% means that the patient never takes the medication and 100% mean that the medication is taken as prescribed at all times. We used PDSQ (Psychiatric Diagnostic Screening Questionnaire) to retrieve several indicators for the mental health of patients diagnosed with RA (major depressive disorder, posttraumatic stress disorder, bulimia/binge-eating disorder, obsessive-compulsive disorder, panic disorder, psychosis, agoraphobia, social phobia, alcohol abuse/dependence, drug abuse/dependence, generalized anxiety disorder, somatization disorder, hypochondriasis).

### 2.1. Description of the Sample of Patients

This study included 119 Romanian patients with RA: 79 patients were treated with conventional drugs (included in Group 1) and 40 patients were treated with biologically injectable treatment (included in Group 2). None of these patients had depression, anxiety or other psychiatric diseases at the time they answered the questionnaires. Each of the 119 patients signed an information form to be included in the present study, as well as an acceptance form for participation. These documents were approved by the Academic and Scientific Ethics and Deontology Committee of the University of Medicine and Pharmacy in Craiova (Registration No. 143/16.06.2017) according to European Union Guidelines (Declaration of Helsinki).

The patient demographic variables contain the following: age, gender, level of education, occupation, as well as type of treatment (conventional or biological). The study was conducted in University Hospital No. 1 from Craiova, Filiasi Municipal Hospital, and Dragasani Municipal Hospital.

The prior treatment or the duration/severity of the disease were neither a required nor an exclusion criterion. The only inclusion criteria were the age (>18 years old), absence of psychiatric health problems and receiving conventional and/or biological therapy as part of management for their RA.

The study took into consideration the disease modifying drugs. Thus, we divided the patients based on the treatments they were receiving into two groups: the group of patients treated with conventional drugs (group 1) and the group of patients treated with biological medicines (group 2). Conventional drugs included conventional disease-modifying antirheumatic drugs (cDMARDs) (Methotrexate, Leflunomide). Biological drugs included Etanercept, Adalimumab, Rituximab, and Certolizumab. All the patients from Group 2 were treated with biologics—combined with synthetic DMARDS or monotherapy. The most used cDMARDs used in combination with biologics, was methotrexate as, there were few patients with adverse events to methotrexate and leflunomide.

### 2.2. Research Tools

The two patient-reported questionnaires (CQR-9 and PDSQ) were answered successively by the 119 patients. Every patient finished responding on the same day to avoid any misalignments within the replies.

PDSQ validation showed it has good test-retest reliability for Romanian translation with an average correlation between 0.63 and 0.88, a mean test-retest of 0.85, for the 13 subscales [28], but a different cut-off score than international limits. Treatment adherence was based on responses to CQR-9 obtained after CQR-19 was optimized [29,30].

### 2.3. Statistical Analysis

Our data was collected using Microsoft Excel and statistically processed using the SPSS v.20 (SPSS Inc., Chicago, IL, USA). The correlation between the subscales of the PDSQ questionnaire and the adherence of each patient was also calculated. We have calculated the parameters of the central trend and dispersion for numerical variables. Moreover, we have analysed the normality using the Kolmogorov–Smirnov test and histograms.

For multivariate analysis, the dependent variable (adherence) and multiple independent variables were involved. For adherence modelling, the listed demographics and PDSQ scales (major depressive disorder, posttraumatic stress disorder, bulimia/binge-eating disorder, obsessive-compulsive disorder, panic disorder, psychosis, agoraphobia, social phobia, alcohol abuse/dependence, drug abuse/dependence, generalized anxiety disorder, somatization disorder, hypochondriasis) were included as independent variables. The variables with a p-value higher than 0.05 were removed from the model. Baseline characteristics, adherence, and PDSQ subscales values were compared between the two groups. T-test and Chi-squared or Fisher’s Exact tests were used for continuous (age, adherence, PDSQ subscales) or categorical (gender, environment, social status, level of education) variables, respectively. The significance level was set at 5%.

## 3. Results

### 3.1. Demographic Characteristics

Patients (n = 119) were divided into two subgroups according to the medical treatment they are currently 79 patients with conventional drugs (group 1) and 40 patients with biological medication (group 2). The socio-demographic data of the patients are reported in Table 2. Age, social environment and employment status were similar (*p* > 0.05), but differences between the two groups were found for gender (*p* = 0.012) and level of education (*p* = 0.002). Participants were mostly women (61% treated with conventional drugs and 83% treated with biological drugs) and had a mean age of 55.85 years, respectively 51.55 years old.

There is a significant association between the gender of the patients and drug treatment; it was found that more women are treated with biological medications than with conventional ones. It was also noted that patients who graduated from University are more likely to be treated with biological therapy than with conventional.

### 3.2. Results from Assessing the Outcomes of CQR-9 Questionnaire

The mean (±SD) adherence value for patients treated with conventional therapy was 60.34 (±12.34), and in patients treated with biological medicines of 81.30 (±13.98), proving lower adherence value for patients in Group 1 compared to Group 2. The 80% threshold shows that patients in group 1 are non-adherent, and group 2 more adherent, as illustrated in Figure 1.

The treatment adherence calculated with the 9 questions optimized questionnaire (CQR-9) for the two groups of patients showed different values as illustrated in Figure 2. The *p*-value < 0.01 (after applying the Kolmogorov–Smirnov test and t-test) demonstrates the statistically significant difference between the therapeutic adherence of patients treated with conventional drugs and those treated with biological therapy.

### 3.3. Results from Assessing the Outcomes from PDSQ Questionnaire

The outcomes for subscales of the PDSQ questionnaire are presented in Table 3 together with the differences between the two groups (after applying the U Mann–Whitney test).

Major depressive disorder and post-traumatic stress disorder were present in the group of patients treated biologically more than in those treated with conventional DMARDs (significant statistical differences with *p* < 0.05), as the clinical picture of symptoms occurred more severely than in patients treated with conventional treatment only. For those in group 2, the awareness of the evolution of the disease with irreversible functional and structural damage could lead to depression.

Moreover, each patient interviewed (from group 2, biological treatment) said during the completing of both questionnaires that the mental state is always influenced by the fluctuations of the disease (felt in the framework of RA). Also, a positive answer to the questions: “Have you felt tired almost every day? Have you frequently come to mind thoughts about a traumatic event?” or “Have you frequently been upset because you’ve thought of a traumatic event?” may indicate a posttraumatic stress disorder (PSD) that the patients feel after a period of exacerbation of the symptoms of the disease.

Bulimia, as a defensive mechanism in the face of depression, by increasing serotonin, occurred significantly statistically different (*p* < 0.05), between the two groups, higher in the group of patients treated with biologics. Only for 3 PDSQ scores: TDM major depression disorder (*p* < 0.01), TSP post-traumatic stress disorder (*p* = 0.013), Bulimia B disorder (*p* = 0.015) we found statistically significant differences between the two groups.

Statistically significant differences between the two groups were not found in any of the other PDSQ scores: obsessive compulsive disorder (*p* = 0.079), panic disorder (*p* = 0.217), psychosis (*p* = 0.217), agoraphobia (*p* = 0,081), social psychosis (*p* = 0.583), alcohol abuse (*p* = 0.660), drug abuse (*p* = 0,131), generalized anxiety disorder (*p* = 0.683), somatization disorder (*p* = 0.411), and hypochondria (*p* = 0.610).

Table 4 shows the number of patients (for each group), which, through their responses, obtained a higher score than the critical point corresponding to each PDSQ subscale.

Two variants are used for critical points: a variant is taken from the user Manual of the PDSQ Questionnaire [31] and a variant adapted to patients in Romania [28]. Following the analysis of critical points (both those set out in the PDSQ manual, as well as in the article) we found differences between the documents. Even if, following the results shown above, there are no notable differences between the two batches, when we analysed each patient in part; we saw differences in the percentage in terms of scores for different subscales.

Nervous bulimia is statistically evidenced with higher value for patients treated with biologics, as patients feel better because therapy is successful.

Alcohol abuse was found in 25 patients (21%), 16 from Group 1 and 9 from Group 2, with no differences between the two groups (*p* = 0.660).

Social phobia has elevated values for both groups, with no statistically significant differences because patients with RA coming out of social circles are excluded or excluded alone (due to the deformations of the joints, etc.). That’s why psychotherapy is recommended for social reintegration.

### 3.4. Modeling Factors Associated with Adherence

Regression analysis of factors associated with treatment adherence highlighted potential associations of data. The Pearson correlation of adherence to the gender, age, and education of patients, as well as their PDSQ scores, is presented in Table 5.

It was noted that socio-demographic factors, gender and education are positively correlated with therapeutic adherence; women and patients with higher education are more adherents.

Major depressive disorder (MDD) and post-traumatic stress disorder (PSD) are positively correlated with therapeutic adherence: 0.317 (*p*-value < 0.01) and 0.225 (*p*-value < 0.05), respectively.

A negative correlation was observed between adherence and panic disorder (r = −0.024), but statistically non important (*p* = 0.396 > 0.05).

No correlation was found between adherence and lifestyle factors: bulimia (Pearson coefficient = 0.072, *p* = 0.218 > 0.05) and alcohol intake (Pearson coefficient = −0.006, *p* = 0.473 > 0.05).

In patients with RA, the adherence prediction model included three variables: age, education and major depressive disorder (calculated with PDSQ, only the 21 questions for MDD). The predictive value for therapeutic adherence, from the estimated regression equation, is obtained with the following formula:ADHERENCE = 43.58 + 0.23 × Age + 5.77 × Education + 1.498 × MDM(1)

The squared sample correlation coefficient, R^2^, was 0.85, that indicated a good predictability ability of this multiple regression model. Using the above formula, we could deduce for each patient with RA, their therapeutic adherence, knowing only information about age, education level, and score obtained following the application of the PDSQ questionnaire for a major depressive disorder (MDM).

## 4. Discussion

The general factors associated with medication adherence in patients with RA are social-economic, disease, therapy, multiple comorbidities, and patient–provider relationship [13,32,33]. The issue of medication adherence is a growing concern not only for RA patients. To address this issue, we conducted a study that examined adherence of RA patients treated with cDMARDs or biologics, combined with synthetic DMARDS, or monotherapy. Firstly, we found that adherence of patients treated with cDMARDs was not optimal (mean = 60.34), whereas the adherence of patients treated with biologics was greater than 80 (mean = 81.3).

Adherence value for RA patients treated with biologic DMARDs was higher compared with those treated with cDMARDs (*p* < 0.01). One explanation for this difference may be the administration route. Mena-Vazquez et al. found that from the RA patients treated with biologic DMARDs, the intravenous route was associated with an almost fourfold greater risk of adherence than the subcutaneous route [34]. From Group 2, some of the patients (the ones with rituximab) had intravenous biologic DMARD infusion, and they were maybe better controlled than others—patients with intravenous treatment had to come to the clinic in order to have the treatment. Moreover, there is a belief among patients that more expensive drugs are more effective, even if the treatment is entirely reimbursed, all of the patients knew the price of the treatment they were taking. Another explanation of the higher adherence is that the patients in the biologic group were already on other therapies and the combination of biologic and cDMARDs is certainly more effective.

Another explanation is linked with the positive correlation between medication adherence and MDD and PSD. A higher proportion of MDD and PSD was observed in Group 2 compared with Group 1. The higher proportion in the biologics group might be related to the longer disease duration in those patients. In Romania, the protocol for RA urges for the use of at least 2 synthetic DMARDs in maximal dosage, for at least 3 months each. In addition, previous protocol specified that, before biologic treatment, the patient should have taken the two synthetic DMARDs combined for another three months. This infers that, patients in this group, which where secondary insufficient responders to many drugs, were having major depressive disorder and post-traumatic stress disorder in a higher proportion.

Following the results, we could say that the two types of treatments (conventional and biological) have differences between adherence scores, MDD and PSD scores, age, and education level.

Depression and anxiety are most often associated with increased functional disability, especially for patients diagnosed with RA. Thus, a reduction in the ability to perform various activities is followed by an increase of at least twice the grade of depression in the following year. Furthermore, it has been shown that reducing disability leads to relief of depression [35], but this is not enough because both depression and anxiety arise from a cumulating of factors such as: social factors, economic factors, age, education etc.

Also, depression remains largely unknown and undertreated in patients suffering from RA. This is largely due to a trend of specialists to focus only on the physical aspects of the disease associated with limited resources. The problem of lack of treatment of depression in patients with RA or other rheumatic conditions is exacerbated by the mistaken conception that it is understandable, occurring secondary to pain and disability, and treatment for depression is not appropriate or necessary. Moreover, the diagnosis of depression is complex because there is an overlap in symptoms (e.g., fatigue, weight loss, insomnia, lack of appetite, etc.) and therefore depression becomes frequently unrecognized [36].

The use of standardized psychiatric questionnaires (which strictly measure depression or anxiety of patients) tends to overestimate the prevalence of depression, as many of these include somatic symptomatology which can be attributed in particular to patients with RA. A lack of familiarity with antidepressant medications can contribute to the sub treatment of depression. Although there is a wide range of antidepressants currently available to physicians, most of them have not been subjected to assessment of effectiveness by patients suffering from serious physical diseases. Current evidence from studies on patient populations diagnosed with psychiatric diseases and other types of chronic pain indicates that major groups of antidepressants (tricyclic) drugs and specific serotonin reuptake inhibitors (administered in appropriate psychotherapeutic doses) have approximately equal efficacy in the treatment of depression [37]. However, these groups differ from the effectiveness of analgesic, tolerability and profile of drug interactions.

Two sets of factors contributing to the onset of depression for patients with RA have been identified, namely the social context and the stage of the patient’s disease. Low socio-economic status, gender, age, as well as functional limitation were all correlated with depression in people diagnosed with RA; systematic inflammation is also associated with depression [38].

Exploring nonadherence in RA patients using DMARDs, Xia et al. [39] found depression to be associated with medication adherence, alongside education, income, and total number of DMARDs. While Xia showed the negative impact of depression on medication adherence, our study found that depression had a small (Pearson coefficient = 0.317), but significant (*p* < 0.01) impact on medication adherence. It’s true that loss of energy, pessimism and negative cognitive style that characterizes depression diminishes the depressed person’s ability to cope with challenges. However, we must see the difference between the two types of depression: major depressive disorder (with symptoms that can last weeks or months, secondary to a reason, as having a disease, for example) and persistent depression (dysthymia or chronic depression) [40]. In our study, we removed the patients with persistent depression, and we aimed to emphasize whether depression due to having the disease, as secondary depression, increase the adherence or not.

Our results must be understood whilst analysing not only MDD, but also PSD (posttraumatic stress disorder) and GAD (generalized anxiety disorder). The Pearson correlation demonstrated that MDD is secondary to having RA without generalized anxiety disorder that conduct to loss of interest in taking the medication, but contrary, the fear of mobility loss, loss of independence, fear of disability or stiffness, feelings that mobilize the patients to continue treatment and increase the adherence.

The challenge of identifying and managing depression in rheumatology clinics is not insignificant. The recognition of depression among patients with rheumatic conditions can be prevented both by the level of the medical system and by the patients’ level of understanding. Depressive symptoms may appear clinically very similar to the symptoms of RA and when depression is not identified or treated patients may misperceive the source of symptomatology. Regarding each patient, the interval and total of depressive symptoms can be underappreciated. Even when patients with rheumatic conditions recognize the symptoms of depression, they may be reluctant to address the subject to the rheumatologist due to time constraints, lack of continuity of the supplier in an academic training centre or because they do not consider this an important fact.

Depression is certainly a treatable disorder and the effectiveness of pharmacological, psychological, and behavioural treatments for depression are well documented. However, more research is required in order to determine which interventions can be more effective among patients with RA and also other rheumatic conditions. It has been shown that a variety of behavioural factors improve disease management and daily functioning, including education and counselling sessions. The results demonstrated that physical exercises were linked to a significant reduction in symptoms of depression in patients with chronic disease. Researchers found that the highest reduction in depressive symptoms came from patients with elevated depression at baseline. The user manual for PDSQ recommends supporting an interview after completing each questionnaire. So for example, if the answers for the following questions: “In the last two weeks have you felt sad or depressed?” and “Have you felt sad or depressed for the most part of the day, almost every day?” are noted with YES, then we can say that those patients suffer in addition to the existing rheumatic disease with anxiety disorders.

Following the completion of the PDSQ questionnaire, 17 (43%) of the patients treated with biological medicinal products at the University Hospital No. 1 in Craiova (Rheumatology clinic) had a positive answer to the question: “In the last six months you have had pain in different body parts?”.

Kekow et al. [41] said in his study on the symptoms of depression and anxiety in RA patients that symptoms were absent at baseline in diagnosis. Also, fatigue and pain had a significant impact on changes in the depression status but did not influence the state of anxiety.

Recent data suggests that depression in RA patients is common, but also poorly recognized in rheumatology. Furthermore, depressive symptoms have been attributed to proinflammatory cytokines including TNF-alpha. Patients with recently diagnosed RA have been overwhelmed and have tried to cope with the disease, so they can generally present sporadic symptoms of depression and anxiety. After a period of treatment (biologically or conventionally) the symptoms of depression and anxiety improve with the disease [42].

Although the characteristics of rheumatic diseases indicate that they negatively affect the mental state of diagnosed patients, there are studies that state that biological treatment (especially the use of etanercept) results in positive outcomes in terms of depression and anxiety. In a meta-analysis, Kappelmann et al. have shown that inflammatory cytokines can have a key beneficial role in the pathogenesis of depression and anxiety. Chronic autoimmune inflammatory disorders such as RA are more common in women than in men, although gender does not necessarily affect the response to anti-TNF therapeutic medication [43].

Even if depression and anxiety in RA patients have been investigated in previous studies [44], there is only a limited number of studies demonstrating the effect of depression and anxiety on RA patients adherence.

Our study has some limitations. Firstly, more patients should be considered over a longer period of time. Secondly, we did not take into account the treatment the patient had before the actual treatment, the disease severity and the duration of the disease, as clinical characteristics to be included as factors influencing the adherence. Furthermore, patients treated with biologics could have different levels of adherence according to the pharmaceutical forms: subcutaneous or intravenous administration, and this could be a new variable to be analysed in a future predictive models. Despite the factors we did not include in the multiple regression model, the R square value assured us the proposed formula has the potential to identify patients that may experience low adherence.

## 5. Conclusions

Lifestyle factors such as bulimia and alcohol abuse are not correlated with treatment adherence at RA patients. Patients with RA may suffer from a majority of psychiatric diseases, but often remain undiagnosed and untreated. Due to their high prevalence, its’ profound impact on the quality of life of patients and the functional disability that the disease determines, it is necessary to recognize and manage depression in patients with RA. A comprehensive approach to managing the mental and physical needs of patients can help improve the treatment adherence and the quality of their lives. Education for patients to address the misconceptions and the de stigmatisation of depression is also recommended because it influences the level of treatment adherence and, in the end, the success of the treatment.

Emotional health is a key factor in the treatment for RA, as well as in the case of rheumatic diseases. The current study underlined the great psychological effects that have an involvement in medical adherence to rheumatologic treatment. Support and patient education were also important factors to be taken into account when treating RA and implicitly the study on adherence to prescribed treatment. In conclusion, the development of the adherence prediction model identified three factors that could increase the level of adherence: age, education and depressive symptoms. As age and education could not be influenced by the rheumatologists, in order to increase the effectiveness of treatment, there is a constant need to complement this treatment with psychological care.

## Figures and Tables

**Figure 1 medicina-56-00178-f001:**
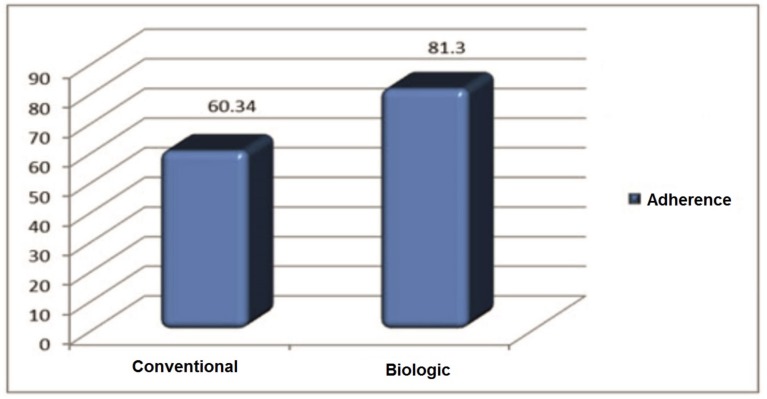
CQR-9 adherence distribution for: (**a**) conventional treatment; (**b**) biological treatment.

**Figure 2 medicina-56-00178-f002:**
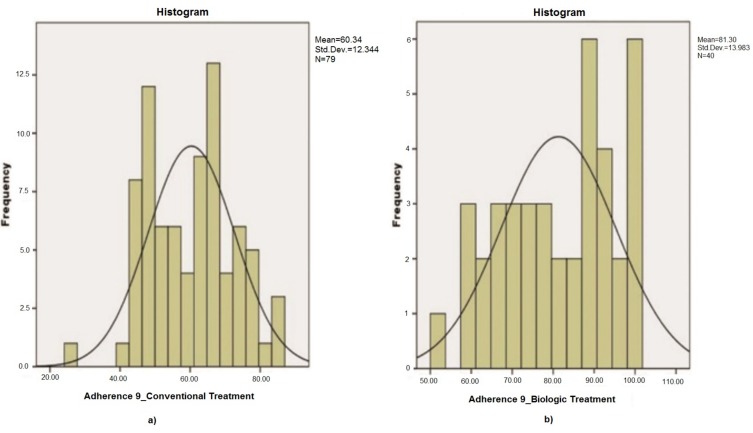
The average level of adherence for the two group.

**Table 1 medicina-56-00178-t001:** CQR-9 (Compliance Questionnaire Rheumatology with 9 items).

	Questions
Q1	I take my anti-rheumatic medicines because I then have fewer problems.
Q2	My medicines are always stored in the same place and that’s why I don’t forget them.
Q3	I take my medicines because I have complete confidence in my rheumatologist.
Q4	If I don’t take my anti-rheumatic medicines regularly, the inflammation returns.
Q5	If I don’t take my anti-rheumatic medicines, my body warns me.
Q6	My health goes above everything else and if I have to take medicines to keep well, I will.
Q7	I use a dose organizer for my medications.
Q8	What the doctor tells me, I hang on to.
Q9	If I don’t take my anti-rheumatic medicines, I have more complaints.

Q = Question. The answers are scored on a 4-point Likert scale with anchors: 1. don’t agree at all; 2. don’t agree; 3. agree; 4. agree very much.

**Table 2 medicina-56-00178-t002:** Demographic characteristic of the two groups of patients.

Characteristics		Group 1 (n = 79)	Group 2 (n = 40)	*p*
		Mean ±SD	Mean ±SD	
Age		55.85 ± 14.4	51.55 ± 13.03	0.267
Age groups	20–50 years	22	16	
	50–60 years	24	15	
	60–70 years	20	8	
	70–80 years	11	0	
	over 80 years old	3	1	
		n (%)	n (%)	
Gender	Female	48 (61%)	33 (83%)	0.012
	Male	31 (39%)	7 (17%)	
Environment	Urban	52 (66%)	33 (83%)	0.085
	Rural	27 (34%)	7 (17%)	
Social status	Employee	42 (53%)	25 (63%)	0.372
	Unemployed	10 (14%)	2 (5%)	
	Pensioner	27 (34%)	13 (33%)	
Level of education	Gymnasium	35 (44%)	4 (10%)	0.002
	High school	16 (20%)	10 (25%)	
	Post-secondary school	8 (10%)	6 (15%)	
	University	20 (25%)	20 (50%)	

n = number of patients, % = percentage, SD = Standard Deviation.

**Table 3 medicina-56-00178-t003:** PDSQ scores for the two groups of patients.

PDSQ Subscales	Group 1 (Mean ± SD)	Group 2 (Mean ± SD)	*p-Value*
Major Depressive Disorder	0.65 ± 1.59	3.95 ± 4.01	<0.01
Posttraumatic Stress Disorder	0.44 ± 1.22	1.25 ± 2.11	0.013
Bulimia	0.27 ± 0.73	1.20 ± 2.31	0.015
Obsessive Compulsive Disorder	0.71 ± 1	0.55 ± 1.34	0.079
Panic disorder	1.52 ± 1.40	1.33 ± 1.77	0.217
Psychosis	0.14 ± 0.47	0.35 ± 0.95	0.135
Agoraphobia	0.62 ± 0.91	0.55 ± 1.48	0.081
Social phobia	1.32 ± 2.25	1.80 ± 3.10	0.583
Alcohol Abuse	0.28 ± 0.64	0.40 ± 0.87	0.660
Drug abuse	0.37 ± 0.86	0.75 ± 1.46	0.131
Generalised Anxiety Disorder	1.33 ± 1.98	1.45 ± 2.32	0.683
Somatisation Disorder	0.65 ± 0.79	0.88 ± 1.11	0.411
Hypochondriasis	0.46 ± 0.75	0.63 ± 1.10	0.610

**Table 4 medicina-56-00178-t004:** Number of patients exceeding the critical (subscale) points of the PDSQ questionnaire.

PDSQ Subscales	Critic Point	Critic Point	Group 1	Group 2
	(Article [30])	(PDSQ Manual [31])	No. Patients with Values over the Critic Point (%)	No. Patients with Values over the Critic Point (%)
Major Depressive Disorder	9	9	0 (0%)	5 (13%)
Posttraumatic Stress Disorder	5	5	1 (1%)	0 (0%)
Bulimia	5	7	0 (0%)	4 (10%)
Obsessive Compulsive Disorder	2	1	15 (19%)	5 (13%)
Panic disorder	4	4	8 (10%)	5 (13%)
Psychosis	1	1	8 (10%)	8 (20%
Agoraphobia	4	4	1 (1%)	2 (5%)
Social phobia	5	4	8 (10%)	7 (18%)
Alcohol Abuse	1	1	16 (20%)	9 (23%)
Drug abuse	2	1	8 (10%)	6 (15%)
Generalized Anxiety Disorder	8	7	1 (1%)	2 (5%)
Somatization Disorder	2	2	11 (14%)	9 (23%)
Hypochondriasis	2	1	6 (8%)	5 (13%)

**Table 5 medicina-56-00178-t005:** The Pearson correlation between adherence and demographic characteristics, respectively PDSQ scores.

	Pearson Correlation (p)
Gender	0.220 (0.016)
Age	−0.006 (0.474)
Education	0.377 (<0.01)
*MDD*	0.317 (<0.01)
*PSD*	0.225 (0.007)
*B*	0.072 (0.218)
*OCD*	−0.055 (0.276)
*PD*	−0.024 (0.396)
*P*	0.135 (0.072)
*A*	0.000 (0.498)
*SP*	0.039 (0.336)
*ALC*	−0.006 (0.473)
*DA*	0.117 (0.102)
*GAD*	0.063 (0.247)
*SD*	0.090 (0.166)
*H*	0.114 (0.108)

MDD = Major Depressive Disorder, PSD = Posttraumatic Stress Disorder, B = Bulimia/Binge-Eating Disorder, OCD = Obsessive-Compulsive Disorder, PD = Panic Disorder, P = Psychosis, A = Agoraphobia, SP = Social Phobia, ALC = Alcohol Abuse/Dependence, DA = Drug Abuse/Dependence, GAD = Generalized Anxiety Disorder, SD = Somatization Disorder, H = Hypochondriasis.

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
