# Peer review of "The Influence of Socio-Demographic Factors, Lifestyle and Psychiatric Indicators on Adherence to Treatment of Patients with Rheumatoid Arthritis: A Cross-Sectional Study"

_medicina, 2020, doi:10.3390/medicina56040178_

Round 1
Reviewer 1 Report
Comments to the Authors
In the present study, the authors aimed to investigate the effects of socio-demographic factors, lifestyle factors, and psychiatric indicators (major depression and post-traumatic stress disorder) on medical adherence in RA patients treated with conventional synthetic DMARDs or biologics-monotherapy. Some comments need to be addressed as follows:
Major comments
- Given that depression and anxiety are the common psychiatric manifestations in RA patients, why the authors choose post-traumatic stress disorder rather than anxiety as the psychiatric indicator?
- Given the possibility of the emergence of immunogenicity (anti-drug antibodies) in RA patients treated with adalimumab, why the authors choose biologics-monotherapy as biologic DMARDs group?
- How to make sure that none of RA patients had depression, anxiety, or the other psychiatric disorders at baseline in the present study?
- It would be better that the authors have a good explanation of a higher adherence value for RA patients treated with biologic DMARDs compare with those treated with csDMARDs.
- How to explain the results that a higher proportion of major depressive disorder and post-traumatic stress disorder was observed in biologics-treated patients compared with csDMARDs-treated patients?
It would be better to have an English editing.

Author Response
Point 1: Given that depression and anxiety are the common psychiatric manifestations in RA patients, why the authors choose post-traumatic stress disorder rather than anxiety as the psychiatric indicator?
Response 1: PDSQ questionnaire assessed Major Depressive Disorder, Posttraumatic Stress Disorder, Generalized Anxiety Disorder, and other 10 psychiatric indicators (Bulimia/Binge-Eating Disorder, Obsessive-Compulsive Disorder, Panic Disorder, Psychosis, Agoraphobia, Social Phobia, Alcohol Abuse/Dependence, Drug Abuse/Dependence, Somatization Disorder, Hypochondriasis). The RA patient could have depressive tendencies due to his diagnosis and because he has fears of mobility loss, disability and loss of independence, he knows what he went through before the RA treatment and he does not want to experience it anymore. We are not talking about anxiety because there is no fear of something for which the patient has a possible treatment. Thus, we thought that posttraumatic stress would be a better marker of the disease’s disability effect on the patient’s psychic. These are the reasons why we choose Posttraumatic Stress Disorder and Generalized Anxiety Disorder rather than only anxiety.
Point 2: Given the possibility of the emergence of immunogenicity (anti-drug antibodies) in RA patients treated with adalimumab, why the authors choose biologics-monotherapy as biologic DMARDs group?
Response 2: You are right and thank you for signalling that error. We are aware of the immunogenicity and actually the regulatory authorities in Romania forbid usage of biologics without a synthetic DMARD, without clear reason (especially RTX). The phrase should have been like this
„All the patients from Group 2 were treated with biologics- combined with synthetic DMARDS or monotherapy. The most used cDMARDs used in combination with biologics, was methotrexate” as, there were few patients with adverse events to MTX and LEF.
We accept the comment and we made the changes according to reality.
Point 3: How to make sure that none of RA patients had depression, anxiety, or the other psychiatric disorders at baseline in the present study?
Response 3: None of RA patients had treatment for depression, anxiety, or other psychiatric disorder at baseline in our study. The possible treatment for these diseases was the real reason we excluded from our study patients with psychiatric problems, we have aimed to assess the patient feedback on healthcare outcomes and patient behaviour without any influence of a psychiatric treatment.
Point 4: It would be better that the authors have a good explanation of a higher adherence value for RA patients treated with biologic DMARDs compare with those treated with csDMARDs.
Response 4: One explanation for this difference may be the administration route. Mena-Vazquez 2017 et al. (Mena-Vazquez, N.; Manrique-Arija, S.; Yunquera-Romero, L.; Urena-Garnica, I.; Rojas-Gimenez, M.; Domic, C.; Jimenez-Nunez, FG.; Fernandez-Nebro, A. Adherence of rheumatoid arthritis patients to biologic disease‑modifying antirheumatic drugs: a cross‑sectional study. Rheumatol Int, 2017, , DOI 10.1007/s00296-017-3758-6) found that from the RA patients treated with bDMARDs, the subcutaneous route was associated with an almost fourfold greater risk of nonadherence than the intravenous route. From the Group 2, some of the patients (the ones with rituximab) had intravenous bDMARD infusion and they were probably better controlled than others - patients with intravenous treatment had to come to the clinic in order to have the treatment. More than this, there is a belief among patients that more expensive drugs are more effective and even if the treatment is entirely reimbursed, all of the patients knew the price of the treatment they were taking. Another explanation of the higher adherence is that the patients in the biologic group were already on other therapies and, for sure, combination of biologic and cDMARDs is more effective.
Other explanation is linked with the positive correlation between medication adherence and MDD (Major Depressive Disorder) and PSD (Posttraumatic Stress Disorder).
Point 5: How to explain the results that a higher proportion of major depressive disorder and post-traumatic stress disorder was observed in biologics-treated patients compared with csDMARDs-treated patients?
Response 5:
The higher proportion in the biologics group might be related to the longer disease duration in those patients. In Romania, the protocol for RA urges for the use of at least 2 synthetic DMARDs in maximal dosage, for at least 3 months each. More than this, previous protocol specified that before biologic treatment, the patient should have taken the two synthetic DMARDs combined for another 3 months. This is why, probably, patients in this group, which where secondary insufficient responders to many drugs, were having major depressive disorder and post-traumatic stress disorder in a higher proportion.
Point 6: It would be better to have an English editing.
An English language editing was used at you suggestion and some improvements were made. See Acknoledgement.

Reviewer 2 Report
The problem addressed in the study, medication adherence, is a very important topic. However, although the used questionnaires were valid I was not able to understand how the model for evaluating adherence was constructed - this has not been opened in the manuscript. Also, I did not understand why age was included in the model? If the aim of the study was to create a model for predicting adherence why were the patients divided in 2 groups based on their DMARD treatment - I find this irrelevant in reference to the aim. Why patients with previous mental health diagnoses were excluded from the study?
Further, to improve readability, I warmly suggest using an English language editing service in the future - I am not quite sure whether I have understood correctly the views and the points the authors are making in the manuscript due to the language constraints.
In addition, I recommend paying attention to scientific writing principles: do not presents results (like no of patients on a certain medication) in methods, do not discuss the results in results section (like "We cannot speak of the pathology of depression, but a tendency towards the awareness 260 of the evolution of the disease with a black perspective on the future.") In the discussion section, please begin with introducing the main results of the study and thereafter discuss them. Of note, it seems that some references are missing in the discussion section. Further, the strengths and weaknesses of the used questionnaires need to be discussed.
Finally, it is a well known fact that education, or health literacy, improves medication adherence. The fact that in this study, major depressive disorder was associated with improved adherence is interesting as on the contrary, previous literature suggests that depression usually increases non-adherence.
In conclusion, I find that significant amount of further work is still required on this manuscript and on the prediction model - I wish the authors luck with this work.
Author Response
Point 1: The problem addressed in the study, medication adherence, is a very important topic. However, although the used questionnaires were valid I was not able to understand how the model for evaluating adherence was constructed - this has not been opened in the manuscript. Also, I did not understand why age was included in the model? If the aim of the study was to create a model for predicting adherence why were the patients divided in 2 groups based on their DMARD treatment - I find this irrelevant in reference to the aim.
Response 1: The questionnaire used for measure the adherence is CQR-9 (see Table 1) that was translated, adapted and optimized prior to this study and the results were published in [31, 32]:
- Subtirelu, M., Turcu-Stiolica, A., Vreju, FA., Ciurea, PL., Dinescu, SC., Barbulescu, AL., Neamtu, J., Danciulescu, RC. Cultural adaptation and optimization of the Compliance Questionnaire-Rheumatology (CQR) through statistical methods for patients with rheumatic diseases. BRAIN-Broad Research in Artificial Intelligence and Neuroscience. 2019, 10, 33-45.
(https://www.edusoft.ro/brain/index.php/brain/article/view/956)
- Subtirelu, M., Turcu-Stiolica, A., Vreju, A., Neamtu, J. Translation and validation of the CQR-19 for Romanian patients with rheumatic diseases. Value Health. 2017, 20(5), A148-A148.
Previous literature showed an important role for the age as a determinant of medication adherence and it is often assumed that older age is related to poorer medication adherence when compared to younger patients. Some studies found no significant relationship between age and medication adherence. We thought why not to include age in the model, especially because the age range is very wide in our study, as you can see in Table 2.
Yes, the aim of the study was to create a model for predicting adherence. The reason we divided in two groups based on their DMARD treatment was the lack of studies that assess the differences in medication adherence by treatment (cDMARDs vs. bDMARDs) for RA patients. We wanted to see all the differences in medication adherence for RA patients. We think there are a lot of differences in terms of adherence between the biologic and synthetic DMARD therapies and we might address those, as well.
Point 2: Further, to improve readability, I warmly suggest using an English language editing service in the future - I am not quite sure whether I have understood correctly the views and the points the authors are making in the manuscript due to the language constraints.
Response 2: An English language editing was used at you suggestion and some improvements were made. See Acknoledgement.
Point 3: In addition, I recommend paying attention to scientific writing principles: do not presents results (like no of patients on a certain medication) in methods, do not discuss the results in results section (like "We cannot speak of the pathology of depression, but a tendency towards the awareness 260 of the evolution of the disease with a black perspective on the future.") In the discussion section, please begin with introducing the main results of the study and thereafter discuss them. Of note, it seems that some references are missing in the discussion section. Further, the strengths and weaknesses of the used questionnaires need to be discussed.
Response 3: We accept the comments and we made the changes according to them.
Point 4: Why patients with previous mental health diagnoses were excluded from the study?
Finally, it is a well-known fact that education, or health literacy, improves medication adherence. The fact that in this study, major depressive disorder was associated with improved adherence is interesting as on the contrary, previous literature suggests that depression usually increases non-adherence.
Response 4: Major depressive disorder (MDD) is a mood disorder that causes a persistent feeling of sadness and loss of interest, affecting the patient’s thinking and behaviour. Most people with depression feel better with medication, psychotherapy or both. These was the reason we excluded from our study patients with psychiatric problems, we have aimed to assess the patient feedback on healthcare outcomes and patient behaviour without any influence of a psychiatric treatment.
Furthermore, the questionnaire PDSQ is a self-administrated patient-reported outcomes that helps a doctor to make an idea of patients thoughts and opinions. And if it exceeds the critical point, the patient will have to make an appointment to see a psychiatrist that have his/her instruments to say if the patient has depression or not. The Pearson correlation between MDD and adherence is small: 0.317, but statistical important (p<0.01). The previous literature suggests that loss of energy, pessimism and negative cognitive style that characterizes depression diminishes the depressed person's ability to cope with challenges. We think we must see the difference between the two types of depression: major depressive disorder (with symptoms that can last weeks or months, secondary to a reason, as having a disease, for example) and persistent depression (dysthymia or chronic depression). In our study, we removed the patients with persistent depression, we aimed to emphasize if depression due to having the disease increase or not the adherence.
Our results should be understood accordingly not only to MDD, but also PSD (Posttraumatic Stress Disorder) and GAD (Generalized Anxiety Disorder). The Pearson correlation demonstrated that MDD is secondary in RA patients, not to generalized anxiety disorder that conduct to loss of interest in taking the medication, but contrary, to the fear of mobility loss, loss of independence, fear of disability or stiffness, feelings that mobilize the patients to continue treatment and increase the adherence.
Please see lines 355-369. We’ve made some clarifications in this direction. Thank you for your suggestions.

Round 2
Reviewer 1 Report
The authors have addressed all my comments
Author Response
Thank you for the review, we appreciate the time that you have taken to review our work.
All the best,
The authors
Reviewer 2 Report
The changes have improved the manuscript.
However, I still have some concerns I hope the authors are able to address.
Comment to response 1:
Please state in the text how and/or why variables were chosen for the model for predicting adherence.
Comment to responses 2-3.
Still at places the text is very difficult to understand due to English language and/or style issues. For example, in the introduction please state the aim of the study so that it is understandable. ("Several mental outpatient settings" means different types of outpatient mental health care and not different types of mental health disturbances. All the mental health problems included in the mental health screening questionnaire don't have to be listed in the introduction - remove this part.) If I understood correctly, there is not much literature on the influence of lifestyle factors and mental health symptoms on RA medication adherence and this study tries to address this lack of knowledge.
Comment to response 4:
I suggest discussing depressive symptoms, not so much depression in the manuscript. You have excluded patients with a diagnosis of depression (and other mental health diagnoses) and you are using a screening questionnaire for mental disorders that cannot be used for actually making a diagnosis. Therefore, your conclusion should be that depressive symptoms, not MDD, are associated with improved adherence to DMARDs.
Patients reporting more depressive symptoms in the bDMARD group were found to be more adherent to the treatment. Do you think that this is generalisable to other setting and patients groups than Romanian RA patients? Or would the more severe RA and associated fatigue be one explanations for these patients scoring higher on MDD questions? This - how you translate the results - should be also discussed in more depth.
Further, major depressive disorder and depression are by definition the same thing. Of course, severe health problems like RA make people more prone to depression - maybe you should consider using the term "secondary depression" in your manuscript? This would maybe be more correct for referring to people who have developed symptoms of depression after RA diagnosis.
Further, factors known to influence medication adherence in general should be mentioned at least briefly either in the introduction or discussion.The limitations of the study:
"Firstly, more patients should be considered over a longer 418 period of time." This was a cross-sectional study, so this is not relevant or maybe state that longitudinal studies are required to confirm our findings or something like that?
"Secondly, we did not take into account the treatment the patient had before the actual 419 treatment, the disease severity and the duration of the disease, as clinical characteristics to be 420 included as factors that could influencinge the adherence. In patients with established disease where conventional medications have not worked, new (biological) strategies may have greater benefits., Treatment costs may be higher but justified by an increase in the quality of life of patients."
Duration of the disease may be one of the factors influencing your results; this is indeed a major limitation. Established disease, treatment cost do not have anything to do with the study limitations so maybe remove from this chapter?
"Furthermore, patients treated with biologicsal could have different levels of adherence according to the pharmaceutical forms: subcutaneous or intravenous administration, and this could be a new variable to be analysed in a future predictive model."
A good point!
"Despite the factors we did notn’t include in the multiple regression model, the R square value assured us the proposed formula has the potential to identify patients that may experience low adherence."
As stated above, is this finding generalisable? This - the generalisability of the results should be discussed in this section.
I suggest also some minor edits/style issues:
Table 2:
Combine age groups 20-50 years? Or present just the mean age +/- SD?
Use male and female
(and to reduce lines possibly Female, n (%) and drop out the line for males? )
Tables 3-4:
These should be combined, there's a lot of repetition in these two tables.
Author Response
Response to Reviewer 2 Comments
Comment to response 1: Please state in the text how and/or why variables were chosen for the model for predicting adherence.
Response1: Thank you for the review, we appreciate the time that you have taken to review our work. Socio-demographic, therapy and patient‐related factors were chosen for the model for predicting adherence. Multivariate analysis involves a dependent variable (adherence) and multiple independent variables (socio-demographic, PDSQ indicators) – see line 170. We have analysed the existing literature about the factors associated with medication adherence of RA patients and why independent variables were chosen in Introduction section.
Comment to responses 2-3.Still at places the text is very difficult to understand due to English language and/or style issues. For example, in the introduction please state the aim of the study so that it is understandable. ("Several mental outpatient settings" means different types of outpatient mental health care and not different types of mental health disturbances. All the mental health problems included in the mental health screening questionnaire don't have to be listed in the introduction - remove this part.) If I understood correctly, there is not much literature on the influence of lifestyle factors and mental health symptoms on RA medication adherence and this study tries to address this lack of knowledge.
Response: Thank you for your recommendation to use an English language editing. Please see the Acknowledgement.
Comment to response 4:
I suggest discussing depressive symptoms, not so much depression in the manuscript. You have excluded patients with a diagnosis of depression (and other mental health diagnoses) and you are using a screening questionnaire for mental disorders that cannot be used for actually making a diagnosis. Therefore, your conclusion should be that depressive symptoms, not MDD, are associated with improved adherence to DMARDs.
Thank you very much for your time and comments, which we think have helped improve the conclusions. Please see line 432.
Patients reporting more depressive symptoms in the bDMARD group were found to be more adherent to the treatment. Do you think that this is generalisable to other setting and patients groups than Romanian RA patients? Or would the more severe RA and associated fatigue be one explanations for these patients scoring higher on MDD questions? This - how you translate the results - should be also discussed in more depth.
We don’t understand your question about Romanian RA patients. We think we have obtained this correlation because the treatment had very good outcomes. In general, a suffering patient looks for a solution and if the treatment makes him feel better, he continues and respects the therapy. Only the treatment inefficiency could contradict this correlation. Patients seem to adhere better when the treatment regimen makes sense to them: when the therapy seems effective, when the benefits seem to exceed the risks/costs (both financial, emotional and physical), and when they feel they have the ability to succeed.
We have discussed the limitations of this study about the severity and the duration of RA, we have supposed that the patients from Group 2 had more severe and longer disease duration because of the described protocol – see line 305. We will analyse in a future study the correlation between the associated fatigue with the MDD score.
Further, major depressive disorder and depression are by definition the same thing. Of course, severe health problems like RA make people more prone to depression - maybe you should consider using the term "secondary depression" in your manuscript? This would maybe be more correct for referring to people who have developed symptoms of depression after RA diagnosis.
The term “secondary depression” is more appropriate and we have used it in the text – see line 353.
Further, factors known to influence medication adherence in general should be mentioned at least briefly either in the introduction or discussion.
We have added this information in the discussion – see line 284
The limitations of the study:
"Firstly, more patients should be considered over a longer 418 period of time." This was a cross-sectional study, so this is not relevant or maybe state that longitudinal studies are required to confirm our findings or something like that?
Yes, a longitudinal study with more patients is required to confirm our findings.
"Secondly, we did not take into account the treatment the patient had before the actual 419 treatment, the disease severity and the duration of the disease, as clinical characteristics to be 420 included as factors that could influencinge the adherence. In patients with established disease where conventional medications have not worked, new (biological) strategies may have greater benefits., Treatment costs may be higher but justified by an increase in the quality of life of patients."
Duration of the disease may be one of the factors influencing your results; this is indeed a major limitation. Established disease, treatment cost do not have anything to do with the study limitations so maybe remove from this chapter?
We have removed this from the discussion of study’s limitations – see line 408.
"Furthermore, patients treated with biologicsal could have different levels of adherence according to the pharmaceutical forms: subcutaneous or intravenous administration, and this could be a new variable to be analysed in a future predictive model."
A good point!
"Despite the factors we did notn’t include in the multiple regression model, the R square value assured us the proposed formula has the potential to identify patients that may experience low adherence."
As stated above, is this finding generalisable? This - the generalisability of the results should be discussed in this section.
Our finding is not generalizable because the study has some limitations, but the formula could identify very well a patient that may experience low adherence because the R square value was very good.
I suggest also some minor edits/style issues:
Table 2:
Combine age groups 20-50 years? Or present just the mean age +/- SD?
We have combined the age groups. See line 190.
Use male and female
(and to reduce lines possibly Female, n (%) and drop out the line for males? )
We think the gender identity could be: Female, Male or Other. A third gender or third sex is a concept in which individuals are categorized, either by themselves or by society, as neither man nor woman. It is also a social category present in societies that recognize three or more genders.
Tables 3-4:
These should be combined, there's a lot of repetition in these two tables.
We could not combine Table 3 and Table 4 because Table 3 presents PDSQ scores for the two groups of patients with p-value from comparing them and Table 4 presents how many patients exceeding the critic Romanian point in every group. The only common information is the column with the Critic point and we have deleted it from Table 3.
Thank you for your time to review and comment. We have considered all your comments and found these useful in improving our manuscript. The amendments are tracked in the attached file.
